# Association of iron homeostasis-related gene polymorphisms with pregnancy and neonatal outcomes in patients with gestational diabetes mellitus

Xiaoli Chen[1‡], Huibin Huang[1‡], Juan Li[1‡], Yansheng Zhang[1], Chenmeng Li[1], Hongbin Xie[2], Lingye Wang[3], Qichang Wu[4]*, Huiming Ye[1,3]*

1 Department of Laboratory Medicine, Fujian Key Clinical Specialty of Laboratory Medicine, Women and Children's Hospital, School of Medicine, Xiamen University, Xiamen, China, 2 Department of Health Management, Women and Children's Hospital, School of Medicine, Xiamen University, Xiamen, China, 3 School of Public Health, Xiamen University, Xiamen, China, 4 Department of Prenatal Diagnosis, Women and Children's Hospital, School of Medicine, Xiamen University, Xiamen, China

‡ XC, HH and JL contributed equally to this work and shared first authorship.
* qichang_wu@163.com (QW); yehuiming@xmu.edu.cn (HY)

**Data Availability Statement:** All relevant data are within the manuscript and its Supporting information files.

## Abstract

### Objective

The purpose of this study was to assess associations between iron homeostasis-related gene polymorphisms and gestational diabetes mellitus (GDM), adverse pregnancy outcomes, and neonatal outcomes.

### Methods

In total, 138 patients with GDM and 74 normal pregnancy controls were recruited. Time-of-flight mass spectrometry was used to genotype single-nucleotide polymorphisms (*H63D* rs1799945, *TMPRSS6* rs855791, *GDF15* rs1059369, rs4808793, *BMP2* rs173107, *C282Y* rs3811647, rs1800562, rs269853, *TF* rs8177240, *TFR2* rs7385804, *FADS2* rs174577, and *CUBN* rs10904850) in 12 candidate genes related to iron homeostasis. Adverse pregnancy outcomes and neonatal health data were collected. Differences in genotype distributions and allele frequencies between patients and controls as well as their correlations with clinical factors were assessed. Additionally, associations between genotype, haemoglobin levels, and ferritin levels were evaluated.

### Results

Pregnant women carrying the *GDF15* rs4808793 allele (C) or *TMPRSS6* rs855791 homozygous mutation (GG) had a significantly higher risk of GDM than that in the control group (p < 0.05). In patients with GDM, the *BMP2* rs173107 heterozygous mutation (AC) was associated with significantly higher haemoglobin levels in late pregnancy compared with those for wild-type (AA) *BMP2* (p < 0.05). Furthermore, in patients with GDM, the *FADS2* rs174577 heterozygous mutation (AC) was associated with a significantly reduced risk of preterm birth

**Funding:** This work was supported by the Major Science and Technology Project of Fujian Provincial Health Commission (2021ZD01006, founded by Xiamen Municipal Health Commission), and the Natural Science Foundation of Xiamen, China (No. 3502Z20227139).

**Competing interests:** The authors have declared that no competing interests exist.

**Abbreviations:** **BMI**, Body Mass Index; **BMP2**, Bone morphogenetic protein 2; **D6D**, Delta-6 desaturase; **EDTA**, Ethylenediaminetetraacetic acid; **FADS2**, Fatty acid desaturase genes 2; **FDR**, False discovery rate; **GDM**, Gestational diabetes mellitus; **HB**, Hemoglobin; **HFE**, Homeostatic Fe regulator; **HOMA-IR**, Homeostasis Model Assessment of Insulin Resistance; **LC-PUFAs**, long-chain polyunsaturated fatty acids; **MAF**, Minor allele frequencies; **NCB**, National Center for Biotechnology Information; **OD**, Optical density value; **OGTT**, Oral glucose tolerance test; **PCR**, Polymerase chain reaction; **SGA**, Full-term small-for-gestational-age; **SNP**, Single nucleotide polymorphism; **SST**, Serum separator tubes; **T2DM**, Type 2 diabetes; **TFR2**, Transferrin receptor 2; **TMPRSS6**, Transmembrane Serine Protease 6; **TTN**, Transient tachypnea of the newborn.

(p < 0.05), the *H63D* rs1799945 heterozygous mutation (CG) was associated with a significantly increased risk of adverse neonatal outcomes (p < 0.05), *TFR2* rs7385804 was associated a significantly reduced probability of caesarean section (p < 0.05), and the G mutation in *TMPRSS6* rs855791 was related to a significantly increased probability of caesarean section (p < 0.05).

## Conclusions

These results suggest that polymorphisms in genes related to iron metabolism could potentially impact pregnancy and neonatal outcomes in patients with GDM. Large-scale studies are needed to further clarify the relationship between these polymorphisms and susceptibility to GDM.

## Introduction

Gestational diabetes mellitus (GDM) refers to impaired glucose tolerance that is first detected or occurs during pregnancy [1]. It is becoming increasingly prevalent worldwide and is one of the most common metabolic disorders during pregnancy [2]. According to the International Diabetes Federation, the incidence of GDM is 14.2% globally and 17.5% in China [3].

GDM not only results in adverse pregnancy outcomes, such as hypertension, polyhydramnios, emergency caesarean section, and premature birth, but also increases the risk of macrosomia and neonatal hypoglycaemia. Furthermore, it can lead to obesity, cardiovascular disease, and type 2 diabetes in offspring. The development of GDM is influenced by genetic factors, inflammatory responses, metabolic abnormalities, and decreased oestrogen receptor expression [4]. The primary pathological mechanisms of GDM are the dysfunction of β cells and insulin resistance.GDM is considered a metabolic disorder that occurs during pregnancy and is thought to be a precursor to type 2 diabetes. It is caused by insulin resistance, which is a result of polygenic factors and pancreatic islet β cell dysfunction [5]. Despite these findings, the cellular mechanism responsible for insulin resistance in GDM is still not fully understood.

Recent studies have shown that abnormal iron metabolism is closely linked to the development of GDM. With respect to the mechanism underlying this connection, tissue iron accumulation can lead to decreased glucose utilisation in muscle tissue, increased glucose output in the liver, insulin resistance, inflammatory responses, and reactive oxygen species production [6–8]. The relationship between the iron status or iron intake and the risk of GDM development has been discussed [9–11]. There is extensive evidence for links between polymorphisms and iron metabolism [12]. However, the exact mechanisms underlying these relationships are still unclear and require further investigation. Additionally, it is important to consider other factors that may contribute to GDM development, such as obesity and gestational weight gain. Hence, a better understanding of the relationship between the iron status, iron intake, and GDM could have important implications for the prevention and management of this condition.

We evaluated the associations between ferritin-associated gene polymorphisms and gestational diabetes by constructing a custom single-nucleotide polymorphism (SNP) array containing 12 polymorphisms in 9 candidate genes with previously demonstrated associations with iron metabolism. We used these polymorphisms to assess the risk of iron deficiency during pregnancy and applied for a patent, which was granted in August 2024 (patent number:

CN202110748981.8). The nine genes were as follows: *H63D* [13], *TMPRSS6* [14], *GDF15* [15], *BMP2* [16], *C282Y* [17, 18], *TF* [19], *TFR2* [19], *FADS2* [19], and *CUBN* [20]. Specific information regarding genetic polymorphisms can be found in S1 Table.

The aim of this study was to analyse candidate genes associated with gestational diabetes using a large-scale array. Correlations between gene polymorphisms related to iron metabolism and various clinical features, pregnancy outcomes, and neonatal outcomes in patients with gestational diabetes were evaluated.

## Materials and methods

### Study participants

This was a retrospective case-control study. From October 2020 to January 2022 (data accessed September 5, 2023), a total of 212 women were recruited from Women and Children's Hospital, School of Medicine, Xiamen University, including 138 patients with GDM and 74 normal pregnancy controls. GDM is diagnosed through a 2-hour, 75g oral glucose tolerance test (OGTT) conducted during mid-pregnancy, specifically between 24 and 28 weeks of gestation. The diagnostic criteria for GDM require at least one abnormal result: a fasting blood glucose level of 5.1 mmol/L after an overnight fast; or, following the consumption of 75g of glucose, a blood glucose level exceeding 10 mM at the one-hour mark or greater than 8.5 mM at the two-hour mark. The control group was obtained from the obstetrics department at the same hospital during the same time period and matched according to the age and parity of the parturients. Only first-time pregnancies and unrelated women were included. Participants with metabolism-related diseases, such as hypertension, diabetes, previous polycystic ovary syndrome, autoimmune diseases, pre-eclampsia, gestational hypertension, and thyroid dysfunction, were excluded. Additionally, women who had alcoholism or multiple pregnancies were not included in the study. The control group consisted of healthy pregnant women without any maternal or foetal disease. One pregnant woman in the normal control group and four pregnant women with diabetes were excluded owing to the choice of other hospitals for delivery, resulting in a lack of relevant data. All participants received iron (150 mg per day) when the concentration of ferritin was less than 30 mg/L. The diagnosis of GDM was determined based on the Guidelines for the Diagnosis and Treatment of Hyperglycemia in Pregnancy (2022) issued by the Chinese Medical Association [21]. Fasting serum and whole blood samples were collected for analyses of ferritin levels, haemoglobin levels, and ferritin-associated gene polymorphisms in the second and third trimesters of pregnancy. This study was conducted in accordance with the ethics guidelines of the Declaration of Helsinki (2002 edition) and was approved by the Ethics Committee of Women and Children's Hospital, School of Medicine, Xiamen University (approval number:KY-2023-067-H01). The content, purpose, execution time, and other aspects this study were explained to participants. All women provided written informed consent to join the study. The study adhered strictly to ethical requirements related to confidentiality.

### Anthropometric, biochemical, and clinical data

During recruitment, age and gestational age (in weeks) were obtained using standard procedures. GDM was diagnosed using the 75-g 2-hr oral glucose tolerance test (OGTT) at 24–28 weeks of pregnancy according to the Chinese Medical Association criteria. A diagnosis of GDM was made if at least one glucose value met the following criteria: fasting plasma glucose $\geq$ 5.1 mmol/L, 1-hour OGTT $\geq$ 10 mmol/L, or 2-hour OGTT $\geq$ 8.5 mmol/L. Serum and whole blood samples were collected from participants in serum separator tubes and ethylenediaminetetraacetic acid (EDTA) tubes and stored at −80˚C until further analyses. The

serum ferritin concentration was determined (Roche Diagnostics kits, Penzberg, Germany) by chemiluminescence (Roche Cobas e801) and the haemoglobin concentration was measured using colorimetry (Mindray BC-5390CRP; Mindray Diagnostic Kit, Shenzhen, China). Clinical data included delivery methods and adverse pregnancy outcomes in newborns, such as macrosomia, neonatal hypoglycaemia, neonatal infection, neonatal hyperbilirubinemia, neonatal vomiting, neonatal shortness of breath (respiratory rate exceeding 60 beats per minute [22]), transient tachypnoea of the newborn (TTN) (typically appearing within the first 2 hours of life in term and late preterm neonates and characterised by tachypnoea and signs of respiratory distress [23]), and small term infants (full-term small-for-gestational-age (SGA) infants, defined as newborns whose birth weight is below the 10th percentile of birth weight or <2 standard deviations from the mean birth weight of the same gestational age and sex [24]).

## DNA isolation and genotyping using matrix-assisted laser desorption/ionisation time-of-flight mass spectrometry (MALDI-ToF-MS)

After thawing the frozen EDTA anticoagulant venous blood at room temperature, genomic DNA was extracted using the Ezup Column Animal Tissue Genomic DNA Extraction Kit (Shenggong Biotechnology Co., Ltd., Qingdao, China), and the specific steps followed the reagent instructions. A volume of 5 μL of DNA solution was separated by electrophoresis on a 1% agarose gel using 1× TAE buffer solution (at 120–180 V). The DNA sample was analysed for quality and concentration using various methods. A clear band on the gel indicated that the DNA was intact and not degraded, while other visible bands confirmed that the concentration met the requirements for PCR. A spectrophotometer was used to determine the concentration and purity of the sample, with an OD value of 1 μL and a ratio of OD 260/280 between 1.7 and 2.0 indicating good DNA quality. Values below 1.7 indicated protein contamination, while values above 2.0 indicated RNA contamination. Generally, a small amount of protein and RNA contamination does not affect ordinary PCR. The MassARRAY® MALDI-TOF System (Time Flight Mass Spectrometry Biochip System) was used for genotyping; this gene typing detection system was exclusively developed and produced by Sequenom, Inc. (San Diego, CA, USA). It is currently the only device for direct SNP typing detection using the principle of time-of-flight mass spectrometry (ToF-MS). After multiple PCR amplifications, SNP sequence-specific extension primers are added. Single-base extension is performed at the SNP site, and different genotypes are extended with different bases. Then, samples were excited by an instantaneous nanosecond ($10^{-9}$ s) strong laser and separated according to the mass-to-charge ratio in a non-electric field drift region. Different genotypes were distinguished based on the quality of the extended bases and the time to reach the detector in a vacuum tube. Sequenom Genotyping Tools and MassArray Assay Design were used to design PCR amplification primers and single-base extension primers for the test site. The specific steps and primer sequences can be found in the patent (Invention patent number: CN202110748981.8) and S1 Table.

## Statistical analysis

SPSS statistics 23 was used for analyses, and values of $p < 0.05$ were considered statistically significant. Normality was evaluated using the Kolmogorov—Smirnov test. Data following a normal distribution are presented as the mean ± standard deviation. Non-normally distributed econometric data are expressed as the median (M) [quartile (P25–P75)]. Allele frequencies of 12 candidate SNPs were estimated by gene counting. Hardy—Weinberg equilibrium was tested in cases and controls using a chi-square test. Clinical characteristics were compared between GDM and control groups using Student's $t$-tests (normally distributed data) and

**Table 1. Demographic and clinical characteristics of the study subjects.**

| Participant Characteristics | Non-GDM | GDM | p |
|---|---|---|---|
| Number | 74 | 138 | - |
| Age (years) | 31(28–33) | 31.5(29–34) | 0.08[a] |
| ferritin(Mid pregnancy)(ng/m/L) | 27.3(17.38–46.5) | 27.4(17.18–45.6) | 0.83[a] |
| ferritin(Late pregnancy)(ng/m/L) | 28.2(18.68–40.15) | 29.1(18.325–45.125) | 0.64[a] |
| HB(Mid pregnancy)(g/L) | 116.8293±8.47097 | 116.8623±9.61157 | 0.82 |
| HB(Late pregnancy)(g//L) | 121.6351±10.07040 | 121.8551±9.36569 | 0.87 |
| Fasting blood glucose (mmol/L) | 4.49+0.276 | 4.94+0.533 | <0.001 |
| Blood glucose (one hour)(mmol/L) | 7.88+1.259 | 10.22+1.626 | <0.001 |
| Blood glucose (two hour)(mmol/L) | 6.53+0.993 | 8.49+1.776 | <0.001 |
| cesarean section | 20(27%) | 57(41.3%) | 0.03[b] * |
| adverse neonatal outcomes | 6(8.1%) | 28(20.2%) | 0.02[b] * |
| Spontaneous preterm delivery | 1(1.4%) | 6(4.3%) | 0.24[b] |

Values are presented as mean ± SD, median (interquartile range) or n (%); and–, no information available

Non-GDM, normal pregnancy controls; and GDM, gestational diabetes.

adverse neonatal outcomes, macrosomia, neonatal hypoglycemia, neonatal infection, neonatal hyperbilirubinemia, neonatal vomiting, neonatal shortness of breath, TTN, and small term infants

P values, t-test independent variables.

[a] Mann—Whitney U test.

[b] Chi-square test or Fisher's exact test..

*:P<0.05 was considered to be statistically significant

Mann—Whitney U-tests (non-parametrically distributed data). The chi-squared ($\chi^2$) test or Fisher's exact test (frequency < 5) was used to compare the frequencies and genotypes of 12 candidate alleles between the experimental and control groups. Associations of 12 candidate gene polymorphisms with adverse pregnancy and neurological outcomes in the GDM group were evaluated. The Mann—Whitney U test was used to evaluate the correlation between ferritin levels and haemoglobin levels in subjects with different genotypes in the GDM group. The ferritin and haemoglobin levels were measured and expressed as either the numerical difference between the median and quartiles or the mean and standard deviation across groups.

## Results

### Demographic and clinical characteristics of the study subjects

Minor allele frequencies (MAF) for all candidate SNPs in the National Center for Biotechnology Information (NCBI) database were greater than 0.05 (S1 Table).A total of 220 participants were enrolled, including 138 patients with GDM and 74 controls (Table 1, Fig 1). There was no significant difference in age between GDM and control groups (p < 0.05). Additionally, there were no significant differences in ferritin levels and haemoglobin levels between the GDM and control groups. The frequencies of caesarean section, adverse neonatal outcomes (macrosomia, neonatal hypoglycaemia, neonatal infection, neonatal hyperbilirubinemia, neonatal vomiting, neonatal shortness of breath, TTN, and small term infants) in the GDM group were significantly higher than those in the control group (p < 0.05).

### Genotype–phenotype associations and gestational diabetes mellitus

We successfully genotyped 9 genes and 12 polymorphisms. rs1800562 was wild-type (GG) in all 212 specimens; accordingly, only the following 11 polymorphisms were included in

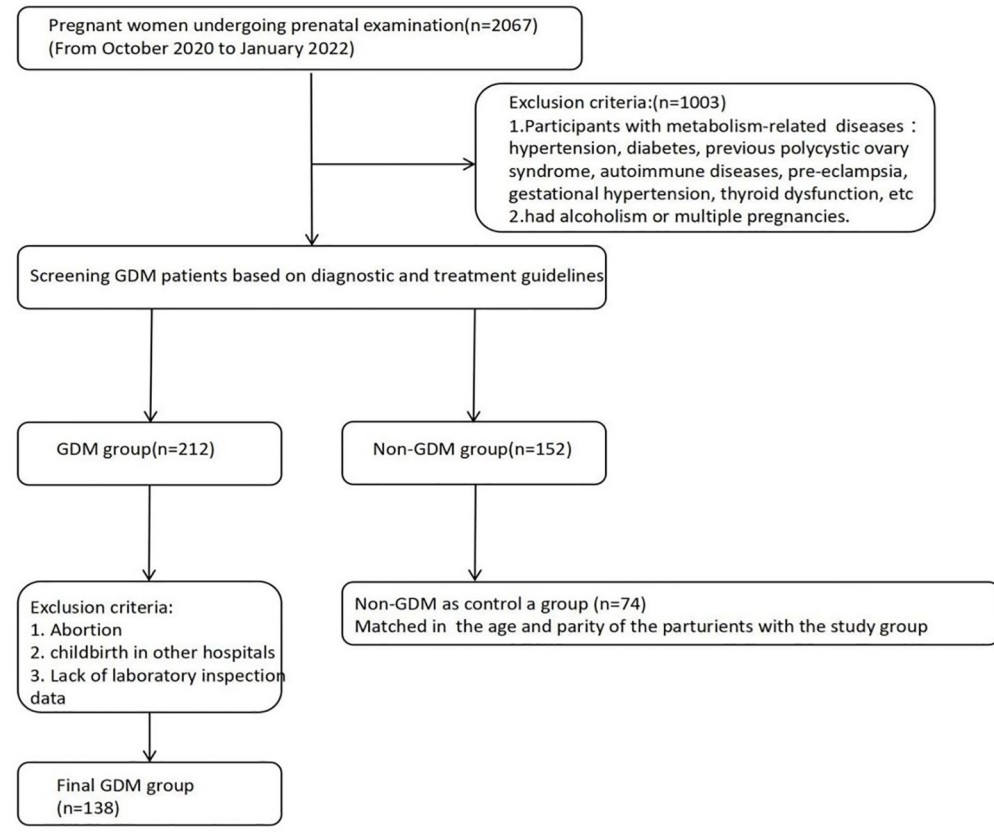

**Fig 1. Patient flow diagram.**

statistical analyses: *H63D* (rs1799945 (C>G)), *TMPRSS6* (rs855791 (A>G)), *GDF15* (rs1059369 (T>A), rs4808793 (G>C)), *BMP2* (rs173107 (A>C)), *C282Y* (rs3811647 (G>A), rs269853 (T>C)), *TF* (rs8177240 (T>G)), *TFR2* (rs7385804 (C>A)), *FADS2* (rs174577 (C>A)), and *CUBN* (rs10904850 (G>A)). Only the CC and CG genotypes were detected for rs1799945. The genotype distributions for all SNPs were in concordance with HWE (p > 0.05) in the control group (Fig 2, S2 Table), indicating that there was no selection bias, population stratification, or genotyping errors [25] in the study population. These results confirmed that the data were representative of the cohort. Additionally, as shown in Fig 2 and S2 Table, the incidence of GDM was higher for *GDF15* rs4808793 allele (C) carriers and *TMPRSS6* rs855791 homozygous mutation (GG) carriers than for normal pregnancy controls (p < 0.05).

## Correlation between the genotype distribution and clinical indicators

As shown in S3 Table, 11 polymorphisms across 9 genes did not exhibit any correlation with ferritin levels during the second and third trimesters. In patients with GDM, carriers of the BMP2 rs173107 heterozygous mutation (AC) had significantly higher haemoglobin levels in late pregnancy than those of patients harbouring the wild type (AA) (p < 0.05) (Fig 3).

## Correlations between the genotype distribution and pregnancy and neonatal outcomes

Among patients with GDM, those carrying the FADS2 rs174577 heterozygous mutation (AC) had a significantly reduced risk of preterm birth (p < 0.05) compared with that for wild-type

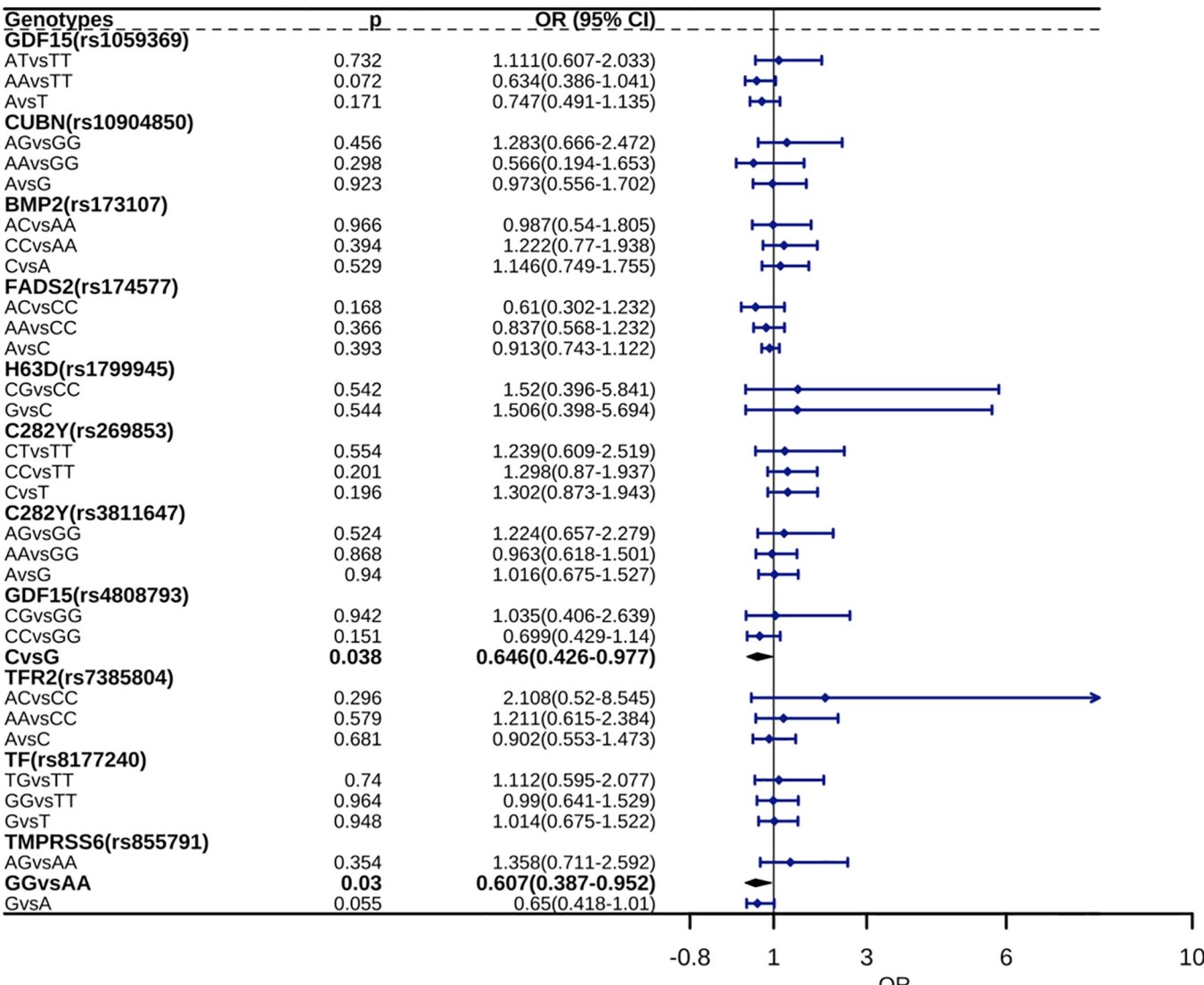

**Fig 2. Associations between genotypes and phenotypes in gestational diabetes mellitus.**

FADS2. The risk of adverse neonatal outcomes (including macrosomia, neonatal hypoglycaemia, neonatal infection, neonatal hyperbilirubinemia, neonatal vomiting, neonatal shortness of breath, TTN, and full-term infants) was significantly increased in individuals carrying the *H63D* rs1799945 heterozygous mutation (CG) (p < 0.05). Patients with GDM carrying the rs7385804 allele of the *TFR2* gene had a significantly reduced probability of caesarean section (p < 0.05), while those carrying the G mutation in the rs855791 allele of the *TMPRSS6* gene had a significantly increased probability of caesarean section (p < 0.05) (Table 2). These results suggest that polymorphisms in iron homeostasis-related genes adversely affect pregnancy or neonatal outcomes in patients with GDM.

## Discussion

The mechanisms underlying GDM remain unclear; although there is mounting evidence linking GDM to various polymorphisms, research on genetic susceptibility to GDM is limited

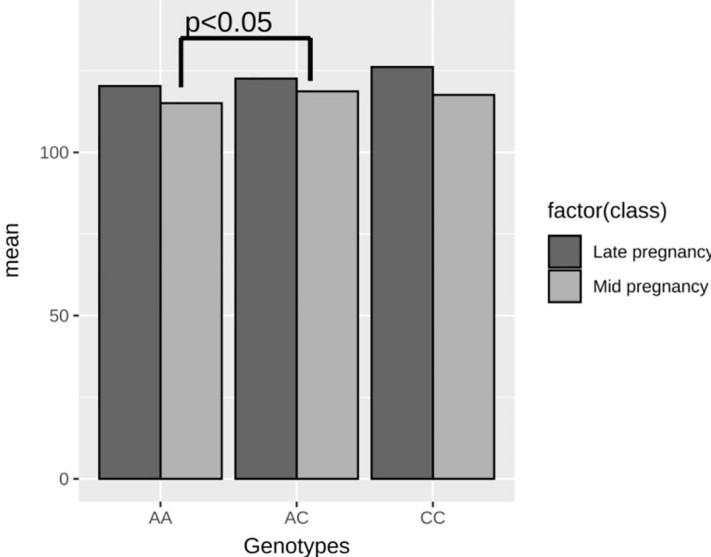

**Fig 3. Correlation between the BMP2 rs173107 polymorphism and different pregnancy stages in patients with GDM.**

[26–28]. Numerous polymorphisms in candidate genes associated with iron nutrition have been identified. Here, we analysed the relationships between candidate genes and the risk of GDM as well as pregnancy and neonatal outcomes. Little research has focused on the associations of candidate genes related to iron nutrition with gestational diabetes. Our study is the first to investigate the link between ferritin-related candidate genes and pregnancy outcomes as well as neonatal outcomes in patients with GDM. These findings underscore the critical need to identify patient subsets base on mutation profiles for monitoring and managing GDM during pregnancy to minimize the risk of adverse outcomes.

In this study, the prevalence of GDM was significantly increased in *GDF-15* rs4808793 allele polymorphism (C) carriers; this finding is consistent with the results of a previous study showing that high GDF-15 levels in late pregnancy are associated with gestational diabetes [29]. Similarly, in this study, we observed that rs855791 homozygous mutation (GG) carriers had a higher probability of developing gestational diabetes than that of normal pregnancy control groups; however, the rs855791 polymorphism was not significantly associated with ferritin levels. Several studies have investigated the relationship between SNPS in the *TMPRSS6* gene and T2DM risk [30, 31]. Liu et al. have shown that there is a significant association between the T allele of *TMPRSS6* rs855791 and an increased risk of GDM [32]. They also proposed that the effect of *TMPRSS6* SNPs on the GDM risk may not be fully explained by the regulation of the iron status in vivo, consistent with the results of this study.

Iron is stored in the human body as ferritin and hemosiderin [26]. The association between excess endogenous and exogenous (supplemental) iron and GDM has been a topic of debate. Several meta-analyses of observational studies have failed to provide conclusive evidence regarding the effect of iron supplementation during pregnancy on the development of GDM [33].

Some studies have suggested that ferritin levels are elevated in GDM, and high levels of haemoglobin in early pregnancy can be an independent risk factor for the development of GDM [34, 35]. Adequate iron is essential for β cell function and glucose homeostasis; however, excessive amounts of iron can cause systemic oxidative stress [36], triggering inflammatory

**Table 2. Correlations between the genotype distribution and pregnancy and neonatal outcomes.**

| gene | variant | Genotype/ allele | Production method | | | | Neonatal outcome | | | | Full term or not | | | |
|---|---|---|---|---|---|---|---|---|---|---|---|---|---|---|
| | | | cesarean birth (n) | natural labour (n) | $\chi^2$ | P | adverse neonatal outcomes (n) | Normal newborn (n) | $\chi^2$ | P | premature birth (n) | full term (n) | $\chi^2$ | P |
| GDF15 | rs1059369 | TT | 21 | 33 | 1.27 | 0.53 | 13 | 41 | 0.93 | 0.63 | 3 | 51 | 0.32 | 0.85 |
| | | AT | 27 | 31 | | | 11 | 47 | | | 2 | 56 | | |
| | | AA | 9 | 17 | | | 4 | 22 | | | 1 | 25 | | |
| | | T | 69 | 97 | 0.00 | 1.00 | 37 | 129 | 0.74 | 0.39 | 8 | 158 | 0.03 | 0.86 |
| | | A | 45 | 65 | | | 19 | 91 | | | 4 | 106 | | |
| CUBN | rs10904850 | GG | 38 | 64 | 2.67 | 0.26 | 21 | 81 | 1.70 | 0.43 | 4 | 98 | 0.71 | 0.70 |
| | | AG | 16 | 14 | | | 7 | 23 | | | 2 | 28 | | |
| | | AA | 3 | 3 | | | 0 | 6 | | | 0 | 6 | | |
| | | G | 92 | 142 | 2.00 | 0.16 | 49 | 185 | 0.18 | 0.67 | 10 | 224 | 0.00 | 1.00 |
| | | A | 22 | 20 | | | 7 | 35 | | | 2 | 40 | | |
| BMP2 | rs173107 | AA | 26 | 40 | 0.95 | 0.62 | 16 | 50 | 2.01 | 0.37 | 3 | 63 | 0.49 | 0.78 |
| | | AC | 24 | 35 | | | 11 | 48 | | | 2 | 57 | | |
| | | CC | 7 | 6 | | | 1 | 12 | | | 1 | 12 | | |
| | | A | 76 | 115 | 0.40 | 0.53 | 43 | 148 | 1.48 | 0.22 | 8 | 183 | 0.00 | 1.00 |
| | | C | 38 | 47 | | | 13 | 72 | | | 4 | 81 | | |
| FADS2 | rs174577 | CC | 13 | 15 | 1.84 | 0.40 | 8 | 20 | 4.86 | 0.09 | 3 | 25 | 4.20 | 0.12 |
| | | AC | 25 | 45 | | | 9 | 61 | | | 1 | 69* | | |
| | | AA | 19 | 21 | | | 11 | 29 | | | 2 | 38 | | |
| | | C | 51 | 75 | 0.02 | 0.89 | 25 | 101 | 0.00 | 0.98 | 7 | 119 | 0.37 | 0.54 |
| | | A | 63 | 87 | | | 31 | 119 | | | 5 | 145 | | |
| H63D | rs1799945 | CC | 56 | 77 | 1.03 | 0.60 | 25 | 108 | 6.04 | 0.05 | 6 | 127 | 8.86 | 0.01 |
| | | CG | 1 | 4 | | | 3 | 2* | | | 0 | 5 | | |
| | | C | 113 | 158 | 0.27 | 0.60 | 53 | 218 | 2.78 | 0.10 | 12 | 259 | 0.00 | 1.00 |
| | | G | 1 | 4 | | | 3 | 2* | | | 0 | 5 | | |
| C282Y | rs269853 | TT | 12 | 26 | 5.05 | 0.08 | 7 | 31 | 0.19 | 0.91 | 2 | 36 | 0.17 | 0.92 |
| | | CT | 35 | 34 | | | 15 | 54 | | | 3 | 66 | | |
| | | CC | 10 | 21 | | | 6 | 25 | | | 1 | 30 | | |
| | | T | 59 | 86 | 0.01 | 0.92 | 29 | 116 | 0.00 | 1.00 | 7 | 138 | 0.01 | 0.91 |
| | | C | 55 | 76 | | | 27 | 104 | | | 5 | 126 | | |
| C282Y | rs3811647 | GG | 25 | 26 | 4.11 | 0.13 | 10 | 41 | 0.14 | 0.93 | 1 | 50 | 1.90 | 0.39 |
| | | AG | 21 | 44 | | | 14 | 51 | | | 3 | 62 | | |
| | | AA | 11 | 11 | | | 4 | 18 | | | 2 | 20 | | |
| | | G | 71 | 96 | 0.14 | 0.70 | 34 | 133 | 0.00 | 1.00 | 5 | 162 | 1.13 | 0.29 |
| | | A | 43 | 66 | | | 22 | 87 | | | 7 | 102 | | |
| GDF15 | rs4808793 | GG | 4 | 9 | 1.44 | 0.49 | 3 | 10 | 0.27 | 0.87 | 1 | 12 | 2.36 | 0.31 |
| | | CG | 23 | 37 | | | 13 | 47 | | | 4 | 56 | | |
| | | CC | 30 | 35 | | | 12 | 53 | | | 1 | 64 | | |
| | | G | 31 | 55 | 1.13 | 0.29 | 19 | 67 | 0.12 | 0.73 | 6 | 80 | 1.26 | 0.26 |
| | | C | 83 | 107 | | | 37 | 153 | | | 6 | 184 | | |
| TFR2 | rs7385804 | CC | 5 | 4 | 4.85 | 0.09 | 1 | 8 | 2.99 | 0.22 | | 9 | 0.51 | 0.78 |
| | | AC | 20 | 17 | | | 11 | 26 | | | 2 | 35 | | |
| | | AA | 32 | 60 | | | 16 | 76 | | | 4 | 88 | | |
| | | C | 30 | 25 | 4.31 | 0.04 | 13 | 42 | 0.25 | 0.62 | 2 | 53 | 0.00 | 1.00 |
| | | A | 84 | 137* | | | 43 | 178 | | | 10 | 211 | | |

(*Continued*)

**Table 2.** (Continued)

| gene | variant | Genotype/allele | Production method | | | | Neonatal outcome | | | | Full term or not | | | |
|---|---|---|---|---|---|---|---|---|---|---|---|---|---|---|
| | | | cesarean birth (n) | natural labour (n) | $\chi^2$ | P | adverse neonatal outcomes (n) | Normal newborn (n) | $\chi^2$ | P | premature birth (n) | full term (n) | $\chi^2$ | P |
| TF | rs8177240 | TT | 24 | 25 | 3.86 | 0.15 | 10 | 39 | 0.08 | 0.96 | 1 | 48 | 1.82 | 0.40 |
| | | TG | 22 | 45 | | | 14 | 53 | | | 3 | 64 | | |
| | | GG | 11 | 11 | | | 4 | 18 | | | 2 | 20 | | |
| | | T | 70 | 95 | 0.11 | 0.74 | 34 | 131 | 0.00 | 0.99 | 5 | 160 | 1.02 | 0.31 |
| | | G | 44 | 67 | | | 22 | 89 | | | 7 | 104 | | |
| TMPRSS6 | rs855791 | AA | 25 | 48 | 3.33 | 0.19 | 14 | 59 | 1.81 | 0.40 | 4 | 69 | 0.48 | 0.79 |
| | | AG | 15 | 17 | | | 9 | 23 | | | 1 | 31 | | |
| | | GG | 17 | 16 | | | 5 | 28 | | | 1 | 32 | | |
| | | A | 65 | 113 | 4.20 | 0.04 | 37 | 141 | 0.01 | 0.90 | 9 | 169 | 0.22 | 0.64 |
| | | G | 49 | 49* | | | 19 | 79 | | | 3 | 95 | | |

*: P<0.05

processes. This process can lead to decreased insulin secretion by the pancreas, increased insulin resistance, and liver dysfunction [37], which contributes to a decrease in glucose uptake by the muscles and an increase in gluconeogenesis, ultimately resulting in the development of GDM [38]. However, some studies have concluded that there is no significant association between high ferritin levels and the development of gestational diabetes [34, 39]. In this study, we did not detect a correlation between serum ferritin levels and haemoglobin levels during the second and third trimesters of pregnancy and the prevalence of gestational diabetes. Additionally, there was no significant correlation between ferritin-related gene polymorphisms and the incidence of gestational diabetes or ferritin levels during the second and third trimesters of pregnancy. In addition, we observed that rs173107 in pregnant women with GDM was not related to serum ferritin levels in the second and third trimesters of pregnancy but was related to the haemoglobin level (p < 0.05). There are two explanations for this result. First, an accurate biomarker for the iron status is lacking. The most widely studied iron state biomarkers include serum ferritin, soluble transferrin receptors, hepcidin, and C-reactive protein. However, it is important to note that some scholars believe that combining multiple biomarkers for evaluation will lead to more reliable conclusions [40]. Second, during the second and third trimesters of pregnancy, the maternal blood volume increases, thereby diluting serum ferritin, which can affect results of statistical analyses. If possible, serum ferritin and haemoglobin levels should be measured in pregnant women in the first trimester. This will provide a clinical basis for establishing gestational ferritin levels related to the development of gestational diabetes.

Numerous studies have demonstrated that patients diagnosed with GDM are at an elevated risk for adverse pregnancy and neonatal outcomes [41–43]. In particular, patients with GDM have an increased likelihood of undergoing a caesarean section, developing postpartum infections, and delivering abnormal newborns. These findings have been widely recognised within the medical community. Our findings indicate that patients with GDM have a greater likelihood of undergoing a caesarean section and giving birth to abnormal newborns. Previous research has also suggested that the increased incidence of caesarean sections and macrosomia in GDM may be attributed to overnutrition [44].

It is important to discuss the reason why gestational diabetes increases the risk of adverse outcomes from the perspective of gene polymorphisms. Research has revealed that the *HFE*

H63D variant in infants is linked to a decrease in birth weight [45]. Additionally, human epidemiological studies have found that *HFE* H63D mutations can cause elevated blood lead levels in children [29]. There is a significant positive correlation between maternal blood lead and umbilical cord blood lead, confirming the existence of maternal embryo lead transport [46]. Increased blood lead levels in pregnant women can lead to an increased incidence of neonatal birth defects and infant developmental abnormalities. High intrauterine lead levels can lead to foetal growth retardation, resulting in neurological defects in newborns. It is worth noting that the effects on neurobehavioral development in newborns are often more pronounced and occur earlier than growth retardation [47]. The H63D allele may contribute to subclinical toxicity by elevating iron levels and lead absorption [48]. This study showed that the rs1799945 polymorphism in the *H63D* gene among patients with GDM was related to a higher incidence of abnormal newborns. This is likely due to increased blood lead absorption in patients with excessive iron levels, resulting in heightened oxidative damage during foetal development. We speculate that rs1799945 is a risk factor for adverse pregnancy outcomes in patients with GDM, and the results of this study provide a theoretical basis for the research on eugenics and fertility testing.

We also observed that among patients with GDM, those carrying the *FADS2* rs174577 heterozygous mutation (AC) had a significantly reduced risk of preterm birth (p < 0.05). According to Cormier et al., rs482548 mutations in *FADS2* can lead to a significant increase in fasting blood glucose levels in Canadian populations. Conversely, mutations at other SNP sites, such as rs7394871, rs174602, rs174570, and rs7482316, can significantly reduce HOMA-IR [49]. These findings indicate that the association between *FADS2* SNPs and glucose metabolism is influenced by the specific mutation site. Additionally, the rs174577 mutation can lower the likelihood of premature delivery in cases of gestational diabetes, suggesting a protective effect. We discovered that the rs7385804 variant (allele A) is associated with an increased likelihood of cesarean section in patients diagnosed with gestational diabetes mellitus (GDM). Conversely, the rs855791 variant (allele G) appears to reduce the probability of undergoing a cesarean section. Previous studies have shown that iron homeostasis is related to obesity, potentially via obesity-induced inflammation [50]. Jonas et al. investigated the relationship between maternal foetal iron processing, placental iron death, oxidative damage, and stress signal activation with foetal growth and observed that alterations in placental iron homeostasis determine peripheral outcomes associated with GDM and/or maternal obesity [51].Therefore, it is likely that these two gene polymorphisms further affect iron homeostasis in pregnant women with diabetes, potentially altering foetal weight and even leading to malposition, which could affect the delivery process.

Our research had limitations that should be acknowledged. First, it is important to note that the association between candidate genes and gestational diabetes can be influenced by gene-diet-lifestyle interactions. However, we did not collect data on dietary intake, lifestyle factors, or physical activity. Hence, it is crucial to consider factors contributing to the relationship between the candidate genes identified in this study and gestational diabetes. Studies have shown that the correlations between genetic polymorphisms and GDM differ among populations and regions [48]. The microbial compositions in the gut, oral cavity, and vagina as well as in the offspring are all correlated with the development of GDM [52]; however, the role of the microbiota was not considered. Additionally, it should be noted that the body mass index (BMI) of the participants was not collected owing to limited resources. However, the BMI of pregnant women with diabetes may differ from that of non-diabetic pregnant women. Therefore, the inclusion of BMI in the analysis may yield more precise and reliable results.

## Conclusion

In conclusion,this study explored the association between polymorphisms in iron homeosta-related genes and the risk of gestational diabetes mellitus and adverse pregnancy outcomes. The results suggest that different polymorphisms in iron homeosta-related genes may increase the risk of gestational diabetes mellitus; different polymorphisms in iron homeosta-related genes may be associated with adverse pregnancy outcomes. Therefore, clinicians should pay attention to the polymorphisms in iron homeosta-related genes during prenatal examination.

## Supporting information

**S1 Fig. Mass spectrometry result chart.** A1 and B1:Mass spectrometry peak spectrum;A2 and B2:Mass spectrometry clustering diagram.
(PDF)

**S1 Table. Gene polymorphism amplification primer sequence.**
(DOCX)

**S2 Table. Association of gene phenotype with gestational diabetes mellitus.**
(DOCX)

**S3 Table. Correlation of GDM hime gene polymorphisms with ferritin and hemoglobin levels.**
(DOCX)

## Acknowledgments

We would like to thank Editage (www.editage.cn) for English language editing.

## Author Contributions

**Conceptualization:** Xiaoli Chen, Huibin Huang, Juan Li.

**Data curation:** Xiaoli Chen, Juan Li, Lingye Wang.

**Formal analysis:** Xiaoli Chen, Juan Li.

**Funding acquisition:** Juan Li, Huiming Ye.

**Investigation:** Hongbin Xie, Lingye Wang.

**Methodology:** Xiaoli Chen, Huibin Huang, Chenmeng Li, Hongbin Xie.

**Project administration:** Huibin Huang, Yansheng Zhang, Hongbin Xie, Huiming Ye.

**Resources:** Yansheng Zhang.

**Software:** Huibin Huang, Yansheng Zhang.

**Supervision:** Qichang Wu, Huiming Ye.

**Validation:** Chenmeng Li, Huiming Ye.

**Visualization:** Chenmeng Li, Lingye Wang.

**Writing – original draft:** Juan Li, Chenmeng Li, Qichang Wu.

**Writing – review & editing:** Xiaoli Chen, Juan Li, Qichang Wu, Huiming Ye.

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
