## [Decision Letter · Decision Letter 0]

13 Aug 2024

PONE-D-24-28141Association of iron homeostasis gene polymorphisms with pregnancy and neonatal outcomes in patients with gestational diabetes mellitusPLOS ONE

Dear Dr. li,

Thank you for submitting your manuscript to PLOS ONE. After careful consideration, we feel that it has merit but does not fully meet PLOS ONE’s publication criteria as it currently stands. Therefore, we invite you to submit a revised version of the manuscript that addresses the points raised during the review process.

We look forward to receiving your revised manuscript.

Kind regards,

Tanja Grubić Kezele, Ph.D., M.D.

Academic Editor

PLOS ONE

Journal Requirements:

This work was supported by the Major Science and Technology Project of Fujian Provincial Health Commission (2021ZD01006, founded by Xiamen Municipal Health Commission), and the Natural Science Foundation of Xiamen, China (No. 3502Z20227139).

The study was supported by the Major Science and Technology Project of Fujian Provincial Health Commission (2021ZD01006, founded by Xiamen Municipal Health Commission), and the Natural Science Foundation of Xiamen, China (No. 3502Z20227139). The authors thank all women who participated in our study. 

This work was supported by the Major Science and Technology Project of Fujian Provincial Health Commission (2021ZD01006, founded by Xiamen Municipal Health Commission), and the Natural Science Foundation of Xiamen, China (No. 3502Z20227139).

Reviewers' comments:

Reviewer's Responses to Questions

**Comments to the Author**

1. Is the manuscript technically sound, and do the data support the conclusions?

Reviewer #1: Partly

Reviewer #2: Partly

2. Has the statistical analysis been performed appropriately and rigorously? 

Reviewer #1: Yes

Reviewer #2: Yes

3. Have the authors made all data underlying the findings in their manuscript fully available?

Reviewer #1: Yes

Reviewer #2: Yes

4. Is the manuscript presented in an intelligible fashion and written in standard English?

Reviewer #1: Yes

Reviewer #2: No

5. Review Comments to the Author

**Reviewer #1:** Many typo and grammatical errors also found in the text such as line 97, 111, 112, etc.

Names of genes need italicization throughout the document. Please follow the guidelines on formatting the name of genes.

How many participants were in each experiment? For example, in Table 1, the authors stated there were 220 participants, but 138 GDM + 74 Control equals 212. Please create a flow chart to indicate the criteria for involving participants.

For Table 1, authors have to add other GDM characteristics. The units for ferritin and HB should be define.

Line 232, rs1800562 has never mention before.

Table 2, the layout is very confusing. Authors have to redesign the table.

rs1799945 did not GG results in Table 2, 3, and 4.

Table 4, the number of participants was 57. Authors have to explain this number. Also, table layout should be adjusted.

Line 429, authors should write the genotype/allele for rs7385804.

**Reviewer #2:** 1. The manuscript is mostly intelligible and written in Standard English, but there are a few areas where the language could be clearer. Conduct a thorough review for grammatical and typographical errors. Consider simplifying complex sentences to enhance readability, particularly for a broader audience that might not be specialized in this field. Ensure that specific errors are removed or checked again, such as the one in lines 122-123 of the Materials and Methods section: "(data 122 accessed September 5, 2023)..

2. Ensure that the results section of the abstract clearly indicates the statistical significance of the findings.

3. Consider clarifying the specific genes studied in the abstract itself to give readers a clearer understanding from the start.

4. The description of the study population and exclusion criteria is comprehensive, but the rationale behind choosing the specific genes for analysis (H63D, TMPRSS6, etc.) should be explained more clearly. Why were these genes selected, and what prior evidence supports their association with GDM?

5. The section discussing the statistical methods (e.g., chi-square tests, t-tests) is appropriate but could benefit from more detail on how the statistical power was determined, considering the sample size.

6. The results are well-organized, but the findings regarding gene polymorphisms and their impact on GDM and pregnancy outcomes need further clarification. Specifically, the clinical significance of the observed associations should be highlighted.

7. The presentation of data could be improved by including visual aids, such as graphs or charts, to illustrate key findings, particularly those related to gene polymorphism frequencies and their association with GDM.

8. The discussion seems to be more tedious being too long and divided into subheadings. Needs extensive and keen revision. Strengthen the discussion around the statistical significance of the results, particularly for associations that are borderline significant.

9. Ensure all conclusions are clearly tied to the data presented, avoiding over-interpretation of the findings.

10. The manuscript mentions that the study was conducted in accordance with the Declaration of Helsinki and approved by an ethics committee. However, it would be helpful to briefly describe how informed consent was obtained from the participants. Mention how the study adheres to principles of research integrity, especially regarding data handling and participant confidentiality.

11. The references are generally appropriate, but it would strengthen the manuscript to include more recent studies on the association between iron metabolism and GDM, if available.

12. Double-check the consistency of citation formatting throughout the manuscript. The overall quality of the written manuscript should be improved

6. PLOS authors have the option to publish the peer review history of their article (what does this mean?). If published, this will include your full peer review and any attached files.

Reviewer #1: No

Reviewer #2: No

---

## [Author Response · Author response to Decision Letter 0]

25 Sep 2024

September 24, 2024

PLOS ONE

Dear Editors and Reviewers:

Thank you for your letter and comments on our manuscript titled “Association of iron homeostasis gene polymorphisms with pregnancy and neonatal outcomes in patients with gestational diabetes mellitus” (Manuscript ID PONE-D-24-28141). These comments helps us improve our manuscript. Accordingly, we have carefully revised and proofread the manuscript, and the detailed corrections are listed below which we hope to meet with approval. Revosed portion are marked in red in the paper.The main corrections in the paper and the responds to the paper and the responds to the reviewer’s comments are as flowing:

Responses to the Reviewer #1:’s comments:

Comment1:  Many typo and grammatical errors also found in the text such as line 97, 111, 112, etc.

Response1：Thank you for pointing this out, we apologize for the language problems in the original manuscript. The language presentation was improved with assistance from a native English speaker with appropriate research background.Language polishing proof visible in the attachment.

Comment2: Names of genes need italicization throughout the document. Please follow the guidelines on formatting the name of genes

Response2：we are grateful for the suggestion, Names of genes have been updated.

Comment3:How many participants were in each experiment? For example, in Table 1, the authors stated there were 220 participants, but 138 GDM + 74 Control equals 212. Please create a flow chart to indicate the criteria for involving participants.

Response3：Thank you for your constructive comment.There are a total of 212 participants. We have incorporated the flowchart as Figure 1 within the manuscript.

Comment4: For Table 1, authors have to add other GDM characteristics. The units for ferritin and HB should be define.

Response4：Thank you for your valuable suggestion. The mid-term glucose tolerance test level has now been incorporated, as illustrated in Table 1. Additionally, the units for blood glucose and hemoglobin have also been included.

Comment5:Line 232, rs1800562 has never mention before.

Response5：Thank you for pointing this out, the rs1800562 variant was found to be wild-type (GG) in 212 specimens. Therefore, only the following 11 gene polymorphisms were included in the statistical analysis.(page 12,line254:We successfully genotyped 9 genes and 12 polymorphisms. rs1800562 was wild-type (GG) in all 212 specimens; accordingly, only the following 11 polymorphisms were included in statistical analyses: )

Comment6: the layout is very confusing. Authors have to redesign the table

Response6：Thank you for your suggestion, we have tried our best to redesign the table.We have replaced the previous Table 2 with a forest map format and included the original Table 2 in the supplementary materials, now designated as Supplementary Table 2. Statistically significant parameters from the former Table 3 are presented in bar chart form, as illustrated in Figure 3. The original Table 3 has been moved to the supplementary section and is now referred to as Supplementary Table 3. Additionally, we have incorporated new statistics into Table 4.

Comment7: rs1799945 did not GG results in Table 2, 3, and 4.

Response7：We sincerely thank the reviewers for their careful reading. Among the variants studied, rs1799945 was found to exhibit only CC and CG phenotypes; therefore, the experimental results did not encompass the GG genotype of rs1799945.

Comment8: the number of participants was 57. Authors have to explain this number. Also, table layout should be adjusted.

Response8：Thank you for your constructive comment. Table 4 presents a correlation analysis of genotype distribution in relation to pregnancy outcomes and neonatal outcomes among patients with metabolic diabetes mellitus. Consequently, the total number of participants included in Table 4 is 138. Regarding the mode of delivery, our data encompasses both cesarean sections and vaginal deliveries, maintaining a total participant count of 138. We have made every effort to ensure that the table is accurately adjusted.we have incorporated new statistics into Table 4.

Comment9:Line 429, authors should write the genotype/allele for rs7385804

Response9：Thank you for your suggestion, We have completed modifications: (page 20,line 404-407，We discovered that the rs7385804 variant (allele A) is associated with an increased likelihood of cesarean section in patients diagnosed with gestational diabetes mellitus (GDM). Conversely, the rs855791 variant (allele G) appears to reduce the probability of undergoing a cesarean section. )

Responses to the Reviewer #2:’s comments:

Comment1:The manuscript is mostly intelligible and written in Standard English, but there are a few areas where the language could be clearer. Conduct a thorough review for grammatical and typographical errors. Consider simplifying complex sentences to enhance readability, particularly for a broader audience that might not be specialized in this field. Ensure that specific errors are removed or checked again, such as the one in lines 122-123 of the Materials and Methods section: "(data 122 accessed September 5, 2023)..

Response1：Thank you for your suggestion, we feel sorry for our poor writings, The language presentation has been enhanced with the assistance of a native English speaker possessing a relevant research background. Complex sentences have been simplified for clarity. Evidence of language refinement can be found in the attached document. Language polishing proof visible in the attachment.(page6,line134-135:(data accessed September 5, 2023)

Comment2:Ensure that the results section of the abstract clearly indicates the statistical significance of the findings.

Response2：Thank you for your suggestion, We have completed modifications:

(Page3-4,line 64-75:Results: Pregnant women carrying the GDF15 rs4808793 allele (C) or TMPRSS6 rs855791 homozygous mutation (GG) had a significantly higher risk of GDM than that in the control group (p < 0.05). In patients with GDM, the BMP2 rs173107 heterozygous mutation (AC) was associated with significantly higher haemoglobin levels in late pregnancy compared with those for wild-type (AA) BMP2 (p < 0.05). Furthermore, in patients with GDM, the FADS2 rs174577 heterozygous mutation (AC) was associated with a significantly reduced risk of preterm birth (p < 0.05), the H63D rs1799945 heterozygous mutation (CG) was associated with a significantly increased risk of adverse neonatal outcomes (p < 0.05), TFR2 rs7385804 was associated a significantly reduced probability of caesarean section (p < 0.05), and the G mutation in TMPRSS6 rs855791 was related to a significantly increased probability of caesarean section (p < 0.05).)

Comment3:Consider clarifying the specific genes studied in the abstract itself to give readers a clearer understanding from the start.

Response3: Thank you for your suggestion, we have completed modifications: (page3,line55-59:Time-of-flight mass spectrometry was used to genotype single-nucleotide polymorphisms (H63D rs1799945, TMPRSS6 rs855791, GDF15 rs1059369, rs4808793, BMP2 rs173107, C282Y rs3811647, rs1800562, rs269853, TF rs8177240, TFR2 rs7385804, FADS2 rs174577, and CUBN rs10904850) in 12 candidate genes related to iron homeostasis.)

Comment4:The description of the study population and exclusion criteria is comprehensive, but the rationale behind choosing the specific genes for analysis (H63D, TMPRSS6, etc.) should be explained more clearly. Why were these genes selected, and what prior evidence supports their association with GDM?

Response4：Thank you for your constructive comment.There are two reasons for choosing these genes:1. All genes included in this study were identified through genome-wide association analysis data and pertinent literature, which substantiated their relationship with iron metabolism. The sources of these references are cited at the first mention of each gene polymorphism.（page5-6,line122-124,These genes include: H63D [1]、TMPRSS6 [2]、GDF15[3]、BMP2 [4]、C282Y [5, 6]）、TF[7]、TFR2 [7]、FADS2 [7]and CUBN [8], Specific information regarding genetic polymorphisms can be found in Supplementary Table 1.）.Previous studies have demonstrated that gestational diabetes is influenced by iron metabolism. Therefore, this study aims to investigate the correlation between genes related to iron metabolism and gestational diabetes. Our team has applied for a patent based on nucleic acid mass spectrometry detection technology, which evaluates the risk of iron deficiency by analyzing combinations of target gene polymorphisms identified in prior research. The patent was granted in August 2024.(patent number:CN202110748981.8).we have completed modifications:(page5-6,line116-125：We evaluated the associations between ferritin-associated gene polymorphisms and gestational diabetes by constructing a custom single-nucleotide polymorphism (SNP) array containing 12 polymorphisms in 9 candidate genes with previously demonstrated associations with iron metabolism. We used these polymorphisms to assess the risk of iron deficiency during pregnancy and applied for a patent, which was granted in August 2024 (patent number: CN202110748981.8). The nine genes were as follows: H63D [1]、TMPRSS6 [2]、GDF15[3]、BMP2 [4]、C282Y [5, 6]）、TF[7]、TFR2 [7]、FADS2 [7]and CUBN [8], Specific information regarding genetic polymorphisms can be found in Supplementary Table 1.)

Comment5:The section discussing the statistical methods (e.g., chi-square tests, t-tests) is appropriate but could benefit from more detail on how the statistical power was determined, considering the sample size.

Response5：Thank you for your constructive comment.we have completed modifications: (page10,line225-229:Clinical characteristics were compared between GDM and control groups using Student’s t-tests (normally distributed data) and Mann–Whitney U-tests (non-parametrically distributed data). The chi-squared (χ2) test or Fisher’s exact test (frequency < 5) was used to compare the frequencies and genotypes of 12 candidate alleles between the experimental and control groups. )

Comment6:The results are well-organized, but the findings regarding gene polymorphisms and their impact on GDM and pregnancy outcomes need further clarification. Specifically, the clinical significance of the observed associations should be highlighted.

Response6：Thank you for your constructive comment.We have placed particular emphasis on the clinical significance of the observed correlations in the Results section.

Comment7: The presentation of data could be improved by including visual aids, such as graphs or charts, to illustrate key findings, particularly those related to gene polymorphism frequencies and their association with GDM.

Response7：Thank you for your suggestion, we have tried our best to redesign the table.We have replaced the previous Table 2 with a forest map format and included the original Table 2 in the supplementary materials, now designated as Supplementary Table 2. Statistically significant parameters from the former Table 3 are presented in bar chart form, as illustrated in Figure 3. The original Table 3 has been moved to the supplementary section and is now referred to as Supplementary Table 3. Additionally, we have incorporated new statistics into Table 4.

Comment8:The discussion seems to be more tedious being too long and divided into subheadings. Needs extensive and keen revision. Strengthen the discussion around the statistical significance of the results, particularly for associations that are borderline significant.

Response8：Thank you for your constructive comment, we have extensively revised the discussion.

Comment9:Ensure all conclusions are clearly tied to the data presented, avoiding over-interpretation of the findings.

Response9：Thank you for your constructive comment, We have extensively and thoroughly revised the conclusion section.

Comment10:The manuscript mentions that the study was conducted in accordance with the Declaration of Helsinki and approved by an ethics committee. However, it would be helpful to briefly describe how informed consent was obtained from the participants. Mention how the study adheres to principles of research integrity, especially regarding data handling and participant confidentiality.

Response10：Thank you for your constructive comment, we have completed modifications:(page7,line159-165:This study was conducted in accordance with the ethics guidelines of the Declaration of Helsinki (2002 edition) and was approved by the Ethics Committee of Women and Children's Hospital, School of Medicine, Xiamen University (approval number:KY-2023-067-H01). The content, purpose, execution time, and other aspects this study were explained to participants. All women provided written informed consent to join the study. The study adhered strictly to ethical requirements related to confidentiality.)

Comment11: The references are generally appropriate, but it would strengthen the manuscript to include more recent studies on the association between iron metabolism and GDM, if available

Response11：We sincerely appreciate the valuable comments. We have incorporated a recent, high-quality reference to strengthen the manuscript. (Zaugg, 2024 )[9]. (page20,line408-412：Jonas et al. investigated the relationship between maternal foetal iron processing, placental iron death, oxidative damage, and stress signal activation with foetal growth and observed that alterations in placental iron homeostasis determine peripheral outcomes associated with GDM and/or maternal obesity）

Comment12:Double-check the consistency of citation formatting throughout the manuscript. The overall quality of the written manuscript should be improved.

Response12：Thank you for your constructive comment, we meticulously examined and refined the citation format to guarantee consistency and adherence to journal requirements.

References

1. Katsarou MS, Papasavva M, Latsi R, Drakoulis N. Hemochromatosis: Hereditary hemochromatosis and HFE gene. Vitamins and hormones. 2019;110:201-22. Epub 2019/02/26. doi: 10.1016/bs.vh.2019.01.010. PubMed PMID: 30798813.

2. Kiss JE. Laboratory and genetic assessment of iron deficiency in blood donors. Clinics in laboratory medicine. 2015;35(1):73-91. Epub 2015/02/14. doi: 10.1016/j.cll.2014.10.011. PubMed PMID: 25676373; PubMed Central PMCID: PMCPMC4451195.

3. Huang C. Research and significance of iron overload GDF-15 gene polymorphism and serum glycoprotein glycosylation modification in thalassemia. Guangxi Medical University [D]. 2017..DOI:10.7666/d.Y3246115.

4. Zhang C, Huo J, Sun J, Huang J. Regional characteristics analysis of iron nutrition status associated gene polymorphism among ethnic minority students aged 6-16. Chin J Public Health. 2019;35:1477-1481

5. Pu W, Sun J, Huang J, Wang L, Tang Y, Li J.The effects of rs855791 and rs3811647 polymorphisms on serum ferritin and soluble transferrin receptor levels in people aged 8-14 years old. J Hyg Res. 2014;43:900-905

6. McLaren CE, Garner CP, Constantine CC, McLachlan S, Vulpe CD, Snively BM, et al. Genome-wide association study identifies genetic loci associated with iron deficiency. PloS one. 2011;6(3):e17390. Epub 2011/04/13. doi: 10.1371/journal.pone.0017390. PubMed PMID: 21483845; PubMed Central PMCID: PMCPMC3069025.

7. Galesloot TE, Janss LL, Burgess S, Kiemeney LA, den Heijer M, de Graaf J, et al. Iron and hepcidin as risk factors in atherosclerosis: what do the genes say? BMC genetics. 2015;16:79. Epub 2015/07/15. doi: 10.1186/s12863-015-0246-4. PubMed PMID: 26159428; PubMed Central PMCID: PMCPMC4498499.

8. McLaren CE, McLachlan S, Garner CP, Vulpe CD, Gordeuk VR, Eckfeldt JH, et al. Associations between single nucleotide polymorphisms in iron-related genes and iron status in multiethnic populations. PloS one. 2012;7(6):e38339. Epub 2012/07/05. doi: 10.1371/journal.pone.0038339. PubMed PMID: 22761678; PubMed Central PMCID: PMCPMC3382217.

9. Zaugg J, Lopez-Tello J, Musial B, Vaughan OR, Fowden AL, Albrecht C, et al. Obesogenic diet in pregnancy disrupts placental iron handling and ferroptosis and stress signalling in association with fetal growth alterations. Cellular and molecular life sciences : CMLS. 2024;81(1):151. Epub 2024/

---

## [Decision Letter · Decision Letter 1]

2 Oct 2024

Association of iron homeostasis gene polymorphisms with

pregnancy and neonatal outcomes in patients with gestational

diabetes mellitus

PONE-D-24-28141R1

Dear Dr. li,

We’re pleased to inform you that your manuscript has been judged scientifically suitable for publication and will be formally accepted for publication once it meets all outstanding technical requirements.

Kind regards,

Tanja Grubić Kezele, Ph.D., M.D.

Academic Editor

PLOS ONE

Additional Editor Comments (optional):

Reviewers' comments:

Reviewer's Responses to Questions

**Comments to the Author**

1. If the authors have adequately addressed your comments raised in a previous round of review and you feel that this manuscript is now acceptable for publication, you may indicate that here to bypass the “Comments to the Author” section, enter your conflict of interest statement in the “Confidential to Editor” section, and submit your "Accept" recommendation.

Reviewer #1: All comments have been addressed

Reviewer #2: All comments have been addressed

2. Is the manuscript technically sound, and do the data support the conclusions?

Reviewer #1: Yes

Reviewer #2: Yes

3. Has the statistical analysis been performed appropriately and rigorously? 

Reviewer #1: Yes

Reviewer #2: Yes

4. Have the authors made all data underlying the findings in their manuscript fully available?

Reviewer #1: Yes

Reviewer #2: Yes

5. Is the manuscript presented in an intelligible fashion and written in standard English?

Reviewer #1: Yes

Reviewer #2: Yes

6. Review Comments to the Author

Reviewer #1: The authors have completely responded all comments. However, I have one concern regarding Figure 3. From my point of view, there does not appear to be a statistically significant difference between AA and AC in mid-pregnancy based on the bar graph. It would be preferable if the authors could present the result in an alternative way and provide the actual P-value.

Reviewer #2: The authors have responded well to the previous comments, and the manuscript has been significantly improved. It is recommended to perform a final review to ensure language consistency, check the clarity of the figures and references, and then accept the manuscript for publication.

7. PLOS authors have the option to publish the peer review history of their article (what does this mean?). If published, this will include your full peer review and any attached files.

Reviewer #1: No

Reviewer #2: No

---

## [Editor Report · Acceptance letter]

30 Oct 2024

PONE-D-24-28141R1 

PLOS ONE

Dear Dr. li, 

I'm pleased to inform you that your manuscript has been deemed suitable for publication in PLOS ONE. Congratulations! Your manuscript is now being handed over to our production team.

Kind regards, 

on behalf of

Prof. dr. Tanja Grubić Kezele 

Academic Editor

PLOS ONE